# The Effect of Ficin Immobilized on Carboxymethyl Chitosan on Biofilms of Oral Pathogens

**DOI:** 10.3390/ijms242216090

**Published:** 2023-11-08

**Authors:** Diana R. Baidamshina, Elena Yu. Trizna, Svetlana S. Goncharova, Andrey V. Sorokin, Maria S. Lavlinskaya, Anastasia P. Melnik, Leysan F. Gafarova, Maya A. Kharitonova, Olga V. Ostolopovskaya, Valeriy G. Artyukhov, Evgenia A. Sokolova, Marina G. Holyavka, Mikhail I. Bogachev, Airat R. Kayumov, Pavel V. Zelenikhin

**Affiliations:** 1Institute of Fundamental Medicine and Biology, Kazan (Volga Region) Federal University, 420008 Kazan, Russia; dianabaidamshina@yandex.ru (D.R.B.); trizna91@mail.ru (E.Y.T.); amelnik200018@mail.ru (A.P.M.); gafarova.lf@rambler.ru (L.F.G.); maya_kharitonova@mail.ru (M.A.K.); olga-ov.kirill@mail.ru (O.V.O.); zhenya_mic@mail.ru (E.A.S.); kairatr@yandex.ru (A.R.K.); 2Department of Biophysics and Biotechnology, Voronezh State University, 394018 Voronezh, Russia; olshannikovas@gmail.com (S.S.G.); andrew.v.sorokin@gmail.com (A.V.S.); maria.lavlinskaya@gmail.com (M.S.L.); artyukhov@bio.vsu.ru (V.G.A.); holyavka@rambler.ru (M.G.H.); 3Laboratory of Bioresource Potential of Coastal Area, Institute for Advanced Studies, Sevastopol State University, 299053 Sevastopol, Russia; 4Biomedical Engineering Research Centre, St. Petersburg Electrotechnical University, 197022 St. Petersburg, Russia; rogex@yandex.com; 5Interdepartment Research Laboratory, Kazan State Academy of Veterinary Medicine Named after N. E. Bauman, 420029 Kazan, Russia

**Keywords:** carboxymethyl chitosan, ficin, oral microbiota, bacterial biofilms

## Abstract

In the last decade, Ficin, a proteolytic enzyme extracted from the latex sap of the wild fig tree, has been widely investigated as a promising tool for the treatment of microbial biofilms, wound healing, and oral care. Here we report the antibiofilm properties of the enzyme immobilized on soluble carboxymethyl chitosan (CMCh) and CMCh itself. Ficin was immobilized on CMCh with molecular weights of either 200, 350 or 600 kDa. Among them, the carrier with a molecular weight of 200 kDa bound the maximum amount of enzyme, binding up to 49% of the total protein compared to 19–32% of the total protein bound to other CMChs. Treatment with pure CMCh led to the destruction of biofilms formed by *Streptococcus salivarius*, *Streptococcus gordonii*, *Streptococcus mutans*, and *Candida albicans*, while no apparent effect on *Staphylococcus aureus* was observed. A soluble Ficin was less efficient in the destruction of the biofilms formed by *Streptococcus sobrinus* and *S. gordonii*. By contrast, treatment with CMCh200-immobilized Ficin led to a significant reduction of the biofilms of the primary colonizers *S. gordonii* and *S. mutans.* In model biofilms obtained by the inoculation of swabs from teeth of healthy volunteers, the destruction of the biofilm by both soluble and immobilized Ficin was observed, although the degree of the destruction varied between artificial plaque samples. Nevertheless, combined treatment of oral *Streptococci* biofilm by enzyme and chlorhexidine for 3 h led to a significant decrease in the viability of biofilm-embedded cells, compared to solely chlorhexidine application. This suggests that the use of either soluble or immobilized Ficin would allow decreasing the amount and/or concentration of the antiseptics required for oral care or improving the efficiency of oral cavity sanitization.

## 1. Introduction

Oral microbiota represent a complex system consisting of more than 700 species of bacteria, fungi, viruses, protozoa, and archaea [1,2]. Oral microorganisms live on the saliva-covered surface of the teeth, in the anaerobic, nutrient-rich gingival crevice, or on the surface of mucosa. Being attached to the surface, bacteria produce an extracellular polymeric substance (EPS), which further protects them from various negative factors, including the immune system of the host, antibiotics, and disinfectants [3]. Therefore, biofilm formation is one of the main factors of microbial resistance and also represents a huge problem for clinics, leading to difficulties in the treatment of many infectious diseases [4]. Among chronic microbial infections, 65–80% are associated with the formation of biofilms [5]. The most obvious example of biofilms in the oral cavity is plaque [6,7]. It includes several types of microorganisms that in turn complicate its eradication.

The most abundant microorganisms found in oral biofilms belong to the genus *Streptococcus* sp., as well as *Candida* sp., *Lactobacillus* sp., *Actinomyces* sp., and others. Of these, *Streptococcus mutans* is considered the main cariogenic [8]. Any changes in the microbiota structure may lead to the disruption of the mutualistic/symbiotic balance and the potential development of oral cavity diseases [9]. The presence of pathogenic bacteria in the oral cavity is a common cause of chronic oral infections such as caries and periodontitis, as well as infections associated with implants. Dental caries is the most common oral disease affecting the majority of the world’s population [10]. Over the past 25 years, the prevalence of dental caries has remained at the same high level, despite all achievements in oral health care [11]. Furthermore, diseases associated with implants are becoming more challenging in the public health context. Thus, about 65% of people with implants suffer from stomatitis, primarily caused by *Candida albicans* [3]. Opportunistic microorganisms can also cause systemic diseases, including those of the gastrointestinal tract and the cardiovascular system [12]. At the same time, the biofilms of the oral cavity are often a shelter for opportunistic microorganisms and a source of recurrent infections [13,14], making their eradication one of the key challenges in oral care.

One of the effective strategies for the treatment of biofilms is their removal by using enzymes capable of destroying the EPS: the destruction of its components can lead to the loss of the biofilm’s structural integrity [15]. Thus, a number of enzymes, such as proteases, glycoside hydrolases, and DNases have been reported as effective destructors of microbial biofilms. In vitro and in vivo studies have shown the ability of mutanase and dextranase to degrade the polysaccharide component of *S. mutans* biofilms as well as mixed biofilms from oral bacteria [16,17,18]. In addition, some data indicate effective degradation of the extracellular DNA of biofilms formed by different *Streptococci* by DNase 1 [19]. Moreover, the combined use of DNase and proteinase K led to a high degree of degradation of the biofilms of cariogenic bacteria by affecting the integrity of the biofilm but not the membrane of bacterial cells [20]. The use of lysozyme and proteinase K led to the dissemination of the biofilms of cariogenic bacteria as well as the reduced adhesion of bacteria to dental hydroxyapatite in the dental plaque [21,22,23]. The synergistic effect of lysozyme and proteinase K with various antibiotics against cariogenic microorganisms has also been demonstrated [23,24]. The ability of some proteases such as bromelain, actinidin, papain, and trypsin to suppress the formation of dental biofilms, both single-bacterial and of polymicrobial communities, has also been reported [25,26].

Ficin, a sulfhydryl protease isolated from the latex of fig trees, can cleave proteins at the carboxyl side of methionine, lysine, arginine, glycine, serine, threonine, valine, asparagines, alanine, and tyrosine [27]. Further, it has been proven to be a bifunctional enzyme, also exhibiting intrinsic peroxidase-like activity [27]. It has been previously shown that Ficin contributes to the destruction of the biofilms of *Staphylococcus aureus* and *Staphylococcus epidermidis* [28]. It has also been shown that Ficin is able to inhibit the biofilm formation of *S. mutans*, reducing the cariogenic virulence of bacteria, including the production of acids and extracellular matrix [29]. Moreover, the ability of Ficin to inhibit the formation of biofilms and to act on the fungal polymorphism of *C. albicans* was shown [30].

Furthermore, a combination of hydrolytic enzymes with antiseptics significantly increases their efficacy against biofilm-embedded bacteria reducing their required doses and treatment times [28,31,32,33]. Thus, chlorhexidine, which is often used for plaque prevention and eradication, remains a “gold standard” of antiplaque agents and oral biofilm control. However, its use on a daily or long-term basis is not feasible due to possible side effects (tartar formation and tooth surface staining) [8]. Therefore, either finding alternative approaches to oral cavity disease treatment or reducing the concentrations of currently applied agents such as chlorhexidine is of immense importance for the improvement of oral care in the global public health context.

Here, we asked whether carboxymethyl chitosan-immobilized Ficin can be used for oral care to remove plaque and increase the efficiency of antiseptics. Our data show that Ficin, in both soluble and carboxymethyl chitosan-immobilized form, is capable of destroying the biofilm formed by oral bacteria as well as enhancing the efficiency of chlorhexidine against biofilm-embedded bacteria two-fold.

## 2. Results

### 2.1. Immobilization of Ficin on Carboxymethyl Chitosan

Ficin was immobilized on carboxymethyl chitosan (CMCh) with molecular weights of either 200, 350 or 600 kDa as described in the Section 4. Further analysis of the samples that were obtained revealed that carboxymethyl chitosan of 200 kDa binds the maximal amount of the enzyme, up to 49% of the total protein, compared to 19% and 32% of the total protein bound to CMCh with molecular weights of 350 and 600, respectively (Table 1). However, the protein seems to be in a catalytically unfavorable state, since the remaining total activity was only 15%, while it was 2- and 4-fold higher for the enzyme bound with CMCh with higher molecular weights. Furthermore, the immobilization of Ficin on CMCh600 lowered the pH- and temperature-sensitivity of the enzyme (Figure 1).

The analysis of the kinetic properties of the enzyme before and after immobilization revealed that the apparent K_m_ and V_max_ values of the immobilized enzyme decreased compared to the free enzyme (Table 2). Of note, while the immobilization on CMCh200 and CMCh350 did not affect the K_m_, V_max_ and k_cat_ were reduced 3- and 6-fold, respectively, suggesting a low speed of catalysis. By contrast, the immobilization of Ficin on CMCh600 reduced all parameters 1.5-fold, assuming low impact on the catalytic properties of the enzyme.

To gain insight into these effects, desorption of the enzyme from all three carriers has been evaluated. As can be seen from Figure 2, the protein dissociated differently from various carriers. Thus, the highest desorption rate within 48 h occurred for the carrier with a molecular weight of 200 kDa, while the lowest was typical for the CMCh with a molecular weight of 350 kDa. Since the carrier surface features also can affect protein desorption, all three CMChs were analyzed with SEM (Figure 3). The surface morphology of CMCh with a molecular weight of 350 kDa exhibited a porous surface that was less pronounced for 600 kDa CMCh and could be observed for 200 kDa CMCh, respectively. In turn, the surface porosity negatively correlated with the desorption speed, confirming that the structure of the carrier slows down protein release (Figure 2). Since the efficiency of the protein bound to CMCh against biofilm is questionable because of steric hindrances, all further tests were performed on Ficin immobilized on CMCh with MW = 200 kDa. For that, a large-scale immobilization of Ficin on CMCh200 was performed. Finally, a bound protein was 14.1 mg/g of CMCh, assuming that 70 mg/mL of CMCh200 immobilized Ficin corresponds to 1 mg/mL of pure enzyme by total protein.

### 2.2. Anti-Biofilm Properties of Immobilized Ficin

In previous studies, promising results of the destruction of staphylococcal biofilms by Ficin were obtained, with the assumption that this property is driven rather by a desorbed enzyme that penetrates into the biofilm and hydrolyzes the proteins of the matrix than by the immobilized protein [28,34]. Therefore, we tested the ability of soluble and CMCh200-immobilized Ficin to destroy the mature biofilms formed by *S. aureus*, *Streptococci* and *C. albicans*, the most common pathogens occurring in the oral cavity [1,30,35,36]. Microorganisms were grown in 96-well plates for 24 h until biofilms were formed, after which the culture liquid was removed, and a fresh broth with compounds to be tested was added. After 3 h of incubation at 37 °C, the residual biofilm in the wells was assessed by crystal violet staining.

Figure 4 shows the residual biofilm in wells treated with either soluble Ficin, CMCh200-immobilized enzyme, or pure CMCh200 as a percent of biofilm biomass in untreated wells. A significant reduction of the biomass after treatment with either soluble or immobilized Ficin could be observed for the biofilms formed by *S. mutans* and *S. aureus*. In particular, *C. albicans* biofilms were reduced 1.5-fold by the treatment with pure enzyme, although a less pronounced effect has been observed when treated with immobilized Ficin, both compared against the untreated control. However, significant biofilm destruction could be observed when compared to the treatment with carboxymethyl chitosan itself. Unexpectedly, while *S. gordonii* biofilms were insensitive to Ficin, almost a 2-fold reduction of the biomass could be observed after treatment with either a carboxymethyl chitosan-immobilized enzyme or carboxymethyl chitosan itself that could possibly be attributed to the mechanical destruction of the biofilm structure.

Then, to investigate how Ficin affects the biofilm structure, the biofilms of *S. gordonii*, *S. mutans*, *S. aureus*, and *C. albicans* treated with either Ficin, pure CMCh, or CMCh200-immobilized Ficin were visualized with confocal laser scanning microscopy (CLSM). For imaging, biofilms were grown for 24 h in BM broth, then the medium was replaced with a fresh one containing Ficin (500 µg/mL), CMCh, or CMCh200-immobilized Ficin (35 mg/mL) and, after 3 h incubation, the cells were stained with SYTO9 and propidium iodide and analyzed using CLSM (Figure 5).

In the control wells, the 24 h old biofilms of all microorganisms reached 10 µm (Figure 5, control lane). After treatment with soluble Ficin, a significant destruction of the *C. albicans* biofilm was observed, while a less pronounced effect was detected for bacterial biofilms, although the density of cells was considerably decreased. For the quantitative evaluation of biofilm destruction, we counted the number of cells (total, viable, and dead) in each Z-stack of confocal images using in-house developed software BioFilmAnalyzer (version 1.2). Data reflecting the total number of cells per layer are shown in Figure 6. As could be seen from the figure, the amount of bacterial cells significantly decreases in samples treated with both types of the enzyme, furthermore, the highest reduction in cells number occurs after the treatment with CMCh-immobilized Ficin.

Notably, the fractions of dead cells were rather comparable in the control and Ficin-treated wells, assuming no antimicrobial activity of Ficin and thus no evolutionary pressure on bacterial resistance development. By contrast, both pure CMCh and CMCh200-immobilized Ficin led to an increase in the fraction of dead cells, especially for *C. albicans*, apparently suggesting the antimicrobial properties of CMCh with a pronounced effect on yeasts. In addition, both compounds resulted in a significant decrease in *S. gordonii* and *S. aureus* biofilms, and CMCh200-immobilized Ficin was able to reduce the biofilm of *S. mutans.*

### 2.3. Anti-Biofilm Properties of Immobilized Ficin on Mixed Biofilms

The efficiency of various approaches to the treatment of biofilms developed with focus on monocultural biofilms often reduces when dealing with mixed communities [16,37,38]. Therefore, the efficiency of soluble and CMCh200-immobilized Ficin in destroying oral biofilms was assessed on model biofilms grown from tooth swabs obtained from healthy volunteers. First, the microbial composition of biofilms grown from either tooth swabs or saliva samples was assessed. For that, swabs from the teeth surfaces of four individuals were obtained and vortexed together. Next, a fraction was used as an inoculum, while the rest was kept as a tooth plaque sample for metagenomic analysis via 16S rRNA barcoding.

Alternatively, saliva samples from the same volunteers were obtained, mixed and inoculated. 24 h-old biofilms grown from both tooth swabs and saliva inoculation were also subjected to metagenomic analysis.

The microbial composition of the microbiomes of tooth plaque and biofilms grown from both tooth swabs and saliva inoculation are shown in Figure 7. As could be seen from the figure, in in vitro grown biofilm, up to 98% of Phyla belong to *Firmicutes*, with an abundance of *Staphylococci*, *Granulicatelli*, and *Streptococci* (see Table 3) and are characterized by reduced biodiversity (see Table 4).

Although artificially grown oral biofilms consisted mainly of *Staphylococci*, *Granulicatelli*, and *Streptococci* (see Table 3), a significant effect of both forms of the enzyme could be observed on two out of four samples (Figure 8A,D). For the samples depicted in Figure 8B,C the biofilm suppression was either less pronounced or even insignificant.

### 2.4. Increasing the Efficiency of Antimicrobials against Staphylococcal Biofilms by Soluble and Chitosan-Immobilized Ficin

Finally, we tested whether Ficin being applied in either of its forms in combination with antiseptics would enhance the effectiveness of the latter. For that, chlorhexidine at the concentration of 16 μg/mL was added to the mature biofilms in combination with the enzyme in either of its forms, and the residual metabolic activity was assessed after 3 h of treatment (Figure 9). Our results indicate that chlorhexidine itself (16 μg/mL) was inefficient against all biofilms under the tested conditions. In marked contrast, when combined with either soluble or immobilized Ficin, a significant reduction in cell viability in the biofilm could be observed for *S. salivarius*, *S. gordonii*, and *S. mutans*. In the case of *S. aureus* and *C. albicans*, enzymatic treatment itself reduced the residual viability, and thus no significant enhancement of the effect of chlorhexidine could be observed.

## 3. Discussion

Various enzymes have been widely investigated as antibiofilm agents, able to destroy the matrix and thereby increase the efficiency of antimicrobials against biofilm-embedded cells [33,39,40,41,42,43]. Among proteases, plant latex proteases represent a promising class of prospective wound healing agents and enhancers of antibiofilm treatment [44,45]. Biofilm destruction properties in vitro have been reported for Ficin, a nonspecific sulfhydryl protease from Fig tree latex, Papain, a protease Papain from Papaya, and Bromelain from the pineapple against staphylococcal biofilms [26,28,31,46]. To overcome the obvious limitations for the direct application of soluble enzymes like their rapid inactivation and autoproteolysis, these enzymes were immobilized on insoluble chitosans, although the usability of heterologous enzymes as antibiofilm agents has been reported to decrease compared to the soluble enzyme [31,34,46,47,48,49,50].

Here, we report the antibiofilm activity of Ficin immobilized on carboxymethyl chitosan (CMCh) which is both soluble in physiological buffers and has no apparent adhesion to biofilms. Indeed, in biofilm destruction tests, no drastic increase in the apparent biomass has been observed in wells treated with pure CMCh (Figure 4), as it has been reported previously in regard to insoluble chitosans [31,34,46]. Moreover, treatment with pure CMCh led to the destruction of the biofilms formed by *S. salivarius*, *S. gordonii*, and *S. mutans*, apparently because of the mechanical destruction of the biofilm matrix. While CMCh remained ineffective against *S. aureus* and *C. albicans* biofilms, it seems to exhibit moderate antimicrobial activity against yeasts (Figure 5). By contrast, the soluble enzyme was unable to reduce the biofilms of *S. sobrinus* and *S. gordonii*, which could be attributed either to the low concentrations of proteins in the matrix produced by these species or to their more rigid structures. Furthermore, the changes in proteins occurring in the matrix of *S. mutans* biofilm [51] would also affect the efficiency of destruction of the latter with Ficin. Of note, the CLSM analysis confirmed the reduction of total cell numbers in *S. gordonii*, *S. mutans*, *S. aureus*, and *C. albicans* biofilms after 3 h treatment with either pure or immobilized Ficin (Figure 5), assuming that the lack of effects detected by CV-staining could be a consequence of the low sensitivity of the approach. Furthermore, the decrease in biofilm biomass (Figure 1) fits well with changes in residual metabolic activity (Figure 9) which, in turn, is proof of biofilm destruction. On the other hand, the ratio of the residual metabolic activity of the treated biofilms and CV-stain data remains at about 0.7–1.0 (Table 5), suggesting that biofilm treatment rather decreases the biomass than kills the cells.

These data allow for the assumption that the practical effectiveness of the treatment with either soluble or CMCh-immobilized enzyme will depend on the individual composition of the microbiota in the oral cavity and the major bacterial species forming the tooth plaque. Indeed, in the model biofilms obtained by the inoculation of swabs from the teeth of healthy volunteers, a significant destruction of the biofilm was observed only in two out of four cases, and in one case the effect was barely significant (*p* < 0.1) (Figure 7). While in some studies the relevance of such artificial tooth plaque has been reported [52,53,54], drastic differences have been revealed when comparing the microbiome structure in tooth plaque and biofilms grown in vitro by the inoculation of saliva and tooth swabs (Figure 6). Thus, the in vitro grown biofilm consists mainly of *Firmicutes*, such as *Staphylococci*, *Granulicatelli*, and *Streptococci*. Consisting of multiple bacteria in various ratios, the tooth plaque would exhibit unique properties in each case which would, in turn, affect the effectiveness of their enzymatic treatment. Furthermore, carbohydrates, which are not affected by proteolytic enzymes, are also one of the main components of the oral biofilm’s matrix, and their fraction varies depending on the microbial community and the individual properties of the host [55,56]. This allows for the assumption of significant differences in the effect of Ficin on biofilms formed by one or multiple species, as well as in vitro and in vivo effects, thus also representing an obvious limitation of this study. In turn, glucans and fructans may bind proteins and thus facilitate biofilm formation [56,57,58,59]. In this regard, the proteolytic destruction of proteins can serve as a tool for biofilm growth prevention.

A combined treatment of the biofilm by enzymes and antimicrobials and antiseptics has been proposed as an advantageous strategy to combat biofilm-associated infections [28,31,34,37,60,61,62]. In our assay, the joint use of both soluble and CMCh-immobilized Ficin with chlorhexidine demonstrated a promising result. Thus, the treatment of oral *Streptococci* by enzyme and chlorhexidine for 3 h led to a significant reduction in the viability of biofilm-embedded cells, compared to a solely chlorhexidine application. This suggests that the use of either soluble or immobilized Ficin would allow for either decreasing the amount and/or concentration of antiseptics for oral care or improve the efficiency of oral cavity sanitization. With that said, for in vivo treatment, the time of plaque contact with both the enzyme and antiseptic is 1–5 min, which would result in a low immediate result. Nevertheless, regular treatment by wash solutions or toothpaste containing Ficin (and antiseptic) would result in long-term effects.

The question of the optimal molecular weight of chitosans for enzyme immobilization remains discussible. In this work, commercially available chitosans with molecular weights of 200, 350 and 600 kDa were modified to obtain CMCh, although the substitution degree increased from 0.46 to 0.78 for CMCh with MW of 600 and 200 kDa, respectively. This fact could explain the apparently higher capacity of 200 kDa CMCh to form a complex with the enzyme (1.5-times more protein was bound). On the other hand, the desorption rate was also the highest, suggesting that the substitution degree is not the main factor that governs protein holding. Nevertheless, the tighter binding of the enzyme apparently leads to its lower activity (Table 1). Furthermore, the kinetic data suggest that immobilization on CMCh600 allows for the retention of the catalytic efficiency of Ficin, while immobilization on CMCh200 leads to binding more total protein (Table 2). Apparently, tight protein holding limits enzyme flexibility during the catalytic act which, in turn, reduces its release from the carrier (Figure 2). The question whether the CMCh-bound Ficin maintains the enzymatic activity remains open. On the one hand, a higher speed of protein release from the carrier could be more advantageous for its practical application as an antibiofilm agent. On the other hand, the use of high molecular chitosan retains enzymatic activity comparable with soluble proteins. Additionally, immobilization on CMCh600 widens the pH and temperature frames for high enzymatic activity (Figure 1) thus allowing for the suggestion of these preparations in biotechnologies with on-flow conditions.

Aside from the force of the carrier-enzyme interaction, the features of the carrier surface also play significant role in protein immobilization. As could be seen from SEM images of CMChs (Figure 3), porosity increases with the increasing molecular weight of CMChs, which partially correlates with the activity of bound protein and negatively correlates with the amount of total protein and its desorption. The protein may become trapped in the pores, which slows down its desorption. Furthermore, the rate of Ficin release is higher from the complex with a carrier of 200 kDa molecular weight compared to 350 kDa. This can be attributed to the fact that the lower molecular weight carrier swells more quickly in the buffer medium, providing better water diffusion in the polymer matrix and facilitating enzyme release. The results of residual protein content determination correlate with the results of the residual catalytic activity of immobilized Ficin (Figure 2B,C): as the protein content reduction in the complex decreases, its activity decreases, and the catalytic activity of the biocatalyst microenvironment increases. Thus, the choice of a carrier seems to be discussible and depends on the testing conditions.

## 4. Materials and Methods

### 4.1. Synthesis of Carboxymethyl Chitosan

The carboxymethyl chitosan was synthesized from acid-soluble chitosans with molecular weights of 200, 350 and 600 kDa (Bioprogress, Shchelkovo, Moscow region, Russia). Chitosan (3 g) was dispersed in 65 mL of isopropanol, and NaOH was added by dropping during 15 min until Ch:NaOH molar ratio of 1:13. Then, a 7-fold molar excess of monochloroacetic acid solution in ethanol was added by dropping and the mixture was incubated for 12 h at 25 °C. A non-soluble product was filtered, neutralized by acetic acid and after wash by pure ethanol was dried at 25 °C [63]. The yield of CMCh was 79–92%; the substitution degree determined by IR-spectroscopy was 0.46, 0.54 and 0.78 for CMCh with MW of 600, 350 and 200 kDa, respectively [64].

### 4.2. Ficin Immobilization on Carboxymethyl Chitosan, Enzymatic Activity Measurements

The Ficin (Sigma-Aldrich, St. Louis, MO, USA) immobilization on CMCh was carried out as follows: one g of CMCh was mixed with 20 mL of Ficin solution (20 mg/mL) in 0.05 M glycine buffer and incubated for 2 h. Then, the formed complex was washed by dialysis in membranes (MWCO = 25 kDa) against Tris-HCl-buffer (50 mM, pH 7.5) until no absorbance at the wavelength of 280 nm could be detected in the buffer. The protein amount in the obtained complexes was determined by using a Lowry assay. The proteolytic activity was evaluated by measuring the absorbance at 410 nm of fragments of the proteolytic digestion of azocasein (Sigma-Aldrich) as described previously [65] with modifications [34]. The obtained heterologous enzyme samples were lyophilized for storage.

The apparent K_m_ and V_max_ values for both free and CMCh-immobilized Ficin were calculated according to the Michaelis–Menden curve and Lineweaver–Burk double reciprocal models using GraphPad Prism 6.0 software. The enzymatic activity was measured at a substrate concentration range of 0.1–100.0 µM under optimal conditions (50 mM Tris-HCl buffer pH 7.5, 37 °C). The enzyme turnover number k_cat_ was calculated as Y = E_t_ × k_cat_ × X/(K_m_ + X).

### 4.3. Bacterial Strains and Growth Conditions

The following bacteria and yeast were used in the study: *Staphylococcus aureus* subsp. *aureus* ATCC 29213, clinical isolate of *Candida albicans* 4940 obtained from the Kazan Institute of Epidemiology and Microbiology (Kazan, Russia); *Streptococcus mutans*, *Streptococcus sobrinus*, *Streptococcus salivarius*, and *Streptococcus gordonii* were isolated from the tooth plaque of healthy volunteers and identified on a Brucker Biotyper. To obtain model oral microbial consortia in vitro, swabs from the teeth or saliva samples of four healthy volunteers were taken and seeded in nutrient broth. The LB medium supplemented with FBS (5% *v*/*v*) and glucose (2% *w*/*v*) and Sabouraud agar were used for maintaining bacterial strains and *Candida albicans*, respectively. The Basal medium (BM) (peptone 7 g, MgSO_4_ × 7 H_2_O 2.0 g and CaCl_2_ × 2 H_2_O 0.05 g in 1.0 L tap water) supplemented with FBS (5% *v*/*v*) and glucose (2% *w*/*v*) was used for the biofilm formation assays [66,67].

### 4.4. Biofilm Assays

To obtain biofilm, bacteria and yeast were cultivated 24 h in BM broth [66] supplemented with FBS (5% *v*/*v*) and glucose (2% *w*/*v*) in 24-well TC-treated polystyrol plates (1 mL per well) under static conditions. Then, the broth was exchanged for fresh broth supplemented with either soluble or chitosan-immobilized Ficin and chlorhexidine in the indicated concentrations. After 24 h of incubation, the plates were subjected to crystal violet staining [68] or metabolic activity was measured with an MTT-assay. The residual biofilm or metabolic activity was calculated as a percentage of the corresponding features in non-treated wells.

### 4.5. Metagenomics

The extraction and purification of DNA for metagenomic analysis was carried out using the Fast DNA^®^SPIN Kit for Soil (MP Biomedicals, Irvine, CA, USA) and a Fast Prep^®^24 homogenizer (MP Biomedicals, USA) according to the manufacturer’s instructions. A 16S rRNA sequencing library was constructed according to the 16S metagenomics sequencing library preparation protocol (Illumina, San Diego, CA, USA) targeting the V3 and V4 hypervariable regions of the 16S rRNA gene. The initial PCR was performed with template DNA using region-specific primers shown to have compatibility with Illumina index and sequencing adapters (forward primer: 5′-TCGTCGGCAGCGTCAGATGTGTATAAGAGACAGTCGTCGGCAGCGTCAGATGTGTATAAGAGACAGCCTACGGGNGGCWGCAG-3′; reverse primer: 5′-GTCTCGTGGGCTCGGAGATGTGTATAAGAGACAGGTCTCGTGGGCTCGGAGATGTGTATAAGAGACAGGACTACHVGGGTATCTAATCC-3′). After the purification of the PCR products with AMPure XT magnetic beads, the second PCR was performed using primers from a Nextera XT Index Kit (Illumina). Subsequently, purified PCR products were visualized using gel electrophoresis and quantified with a Qubit dsDNA HS Assay Kit (Thermo Scientific, Waltham, MA, USA) on a Qubit 2.0 fluorometer. The sample pool (4 nM) was denatured with 0.2 N NaOH, diluted further to 4 pM, and mixed with 20% (*v*/*v*) denatured 4 pM PhiX, prepared following Illumina guidelines. Sequencing was performed on the Illumina MiSeq platform in 2 × 300 bp mode. Reads were further processed and analyzed using QIIME software, version 1.9.1 [69], according to protocols. Before filtering, there were 26,740–45,375 (median 32,590) read pairs per sample. After quality filtering, chimera filtering, and rarefying, we analyzed on average 16,189 joined read pairs. Sequences were clustered into operational taxonomic units (OTU) based on the 97% identity threshold (open reference-based OTU picking strategy); the SILVA database v.138 [70] was used. To characterize the richness and evenness of the bacterial community, the alpha diversity index was calculated using phylogenetic diversity (PD whole tree), Chao1, Simpson’s and Shannon’s metrics.

### 4.6. Scanning Electron Microscopy

The surface features of chitosans with various MW were assessed with scanning electron microscopy. The samples were coated in vacuum with gold on SCD 004 (Balzers, AG, Balzers, Liechtenstein). SEM was performed with the Quanta 200 microscope (FEI Company, Hillsboro, OR, USA) at 29 kV.

### 4.7. Confocal Laser Scanning Microscopy

The spatial structure of microbial biofilms was assessed with Confocal laser scanning microscopy. Biofilms were grown 24 h in BM broth in cell imaging chambered coverslips with 8 wells (Ibidi, Gräfelfing, Germany) and stained with SYTO 9 (ThermoFisher Scientific) at a final concentration of 0.02 μg/mL (green fluorescence) and propidium iodide (Sigma) at a final concentration of 3 μg/mL (red fluorescence) and analyzed by an Olympus IX83 inverted microscope supplemented with a STEDYCON ultrawide extension platform. The number of non-viable cells was estimated as the relative fraction of the red cells among all cells in each Z-stack by BioFilmAnalyzer software [71] and averaged for whole image.

### 4.8. Statistical Analysis

All experiments were performed in triplicates with 3 technical repeats in each one. Numerical data were compared for statistically significant differences using the Kruskal–Wallis test with a significance threshold at *p* < 0.05. All calculations were performed in GraphPad Prism 6.0 software.

## 5. Conclusions

Taken together, our data suggest that Ficin immobilized on soluble carboxymethyl chitosan could be of considerable interest for oral care, from additives to mouth wash solutions, toothpaste, and gum to antiseptic films widely used in stomatology. While the exposition of plaque to Ficin in wash solutions, toothpaste, gum will be short, the application in antiseptic films could be combined with antiseptics and would have long-term contact with bacterial biofilms. In this study, statistically significant effects could be observed after 3 h, that is too long for the real-life conditions which differ from in vitro conditions and therefore additional assays are required. While the molecules of carboxymethyl chitosan would treat the biofilms of oral *Streptococci* on the one hand, on the other, the fast desorption of the enzyme would lead to the appearance of soluble enzyme affecting the biofilm matrix components formed by other oral flora. Furthermore, the increased efficiency of chlorhexidine combined with either soluble or immobilized Ficin allows for either reducing the amount and/or concentration of the antiseptics required for oral care, or for improving the efficiency of oral cavity sanitization.

## Figures and Tables

**Figure 1 ijms-24-16090-f001:**
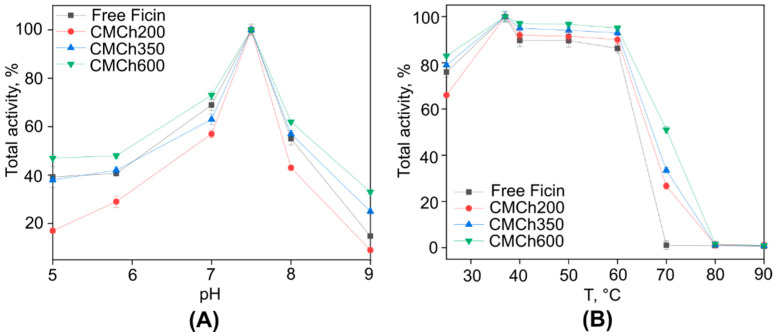
The effect of pH (**A**) and temperature (**B**) on the total activity of soluble Ficin and the enzyme immobilized on CMCh with molecular weights of either 200, 350 or 600 kDa. The protein amount and total activity of the freshly prepared complex was considered as 100%. A mean value ± standard deviation from nine independent measurements is shown.

**Figure 2 ijms-24-16090-f002:**
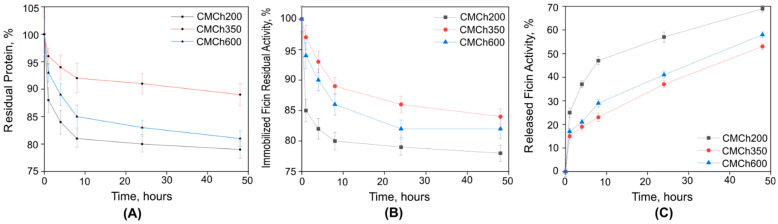
Desorption of the immobilized Ficin from CMCh with molecular weights of either 200, 350 or 600 kDa (**A**) and enzymatic activity of immobilized (**B**) and released (**C**) Ficin in 50 mM Tris-HCl buffer pH 7.5. The protein amount and specific activity of the freshly prepared complex was considered as 100%. A mean value ± standard deviation from nine independent measurements is shown.

**Figure 3 ijms-24-16090-f003:**
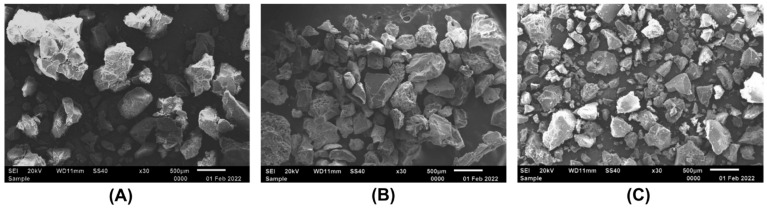
SEM images of surface of carboxymethyl chitosan with molecular weights of 200 kDa (**A**), 350 kDa (**B**) and 600 kDa (**C**). SEM was performed with a Quanta 200 microscope with a magnification of 30×.

**Figure 4 ijms-24-16090-f004:**
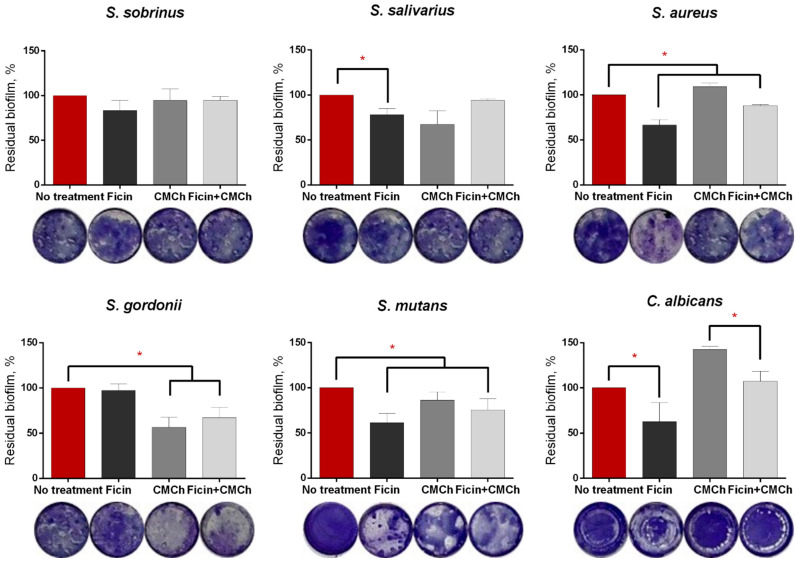
The effect of soluble and carboxymethyl chitosan-immobilized Ficin on monomicrobial biofilms in vitro. 24 h-old biofilms grown in BM broth were gently washed by PBS and loaded with fresh BM broth supplemented with either soluble Ficin (500 µg/mL) or CMCh-immobilized Ficin (35 mg/mL) followed by 3 h incubation and quantification of the residual biofilms by crystal-violet staining. The images of representative stained wells are shown below the bars. Asterisks (*) denote a statistically significant difference with the shown treatment (*p* < 0.05). A mean value ± standard deviation from nine independent measurements is shown.

**Figure 5 ijms-24-16090-f005:**
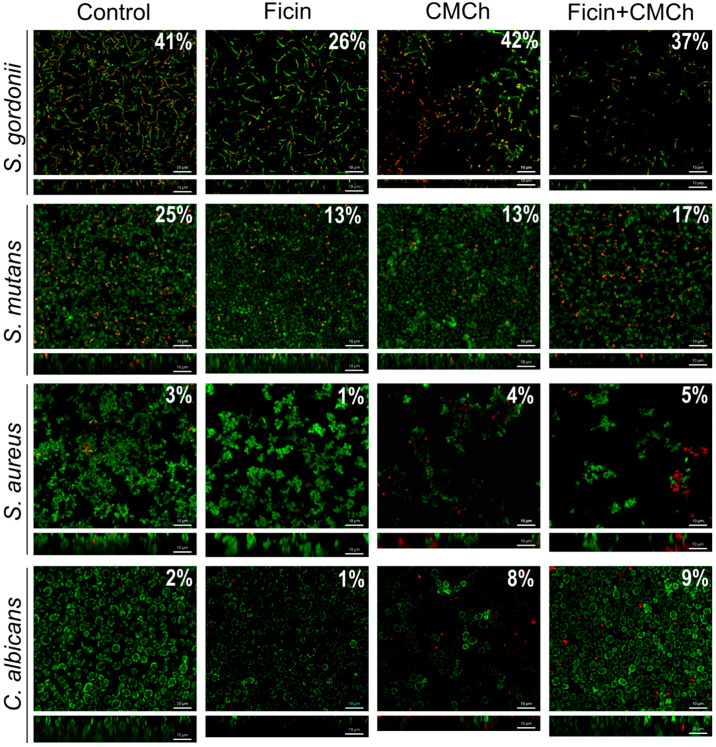
Confocal laser scanning microscopy. Microbial 24 h-old biofilms were treated for 3 h with either Ficin, pure CMCh, or CMCh200-immobilized Ficin. Then, cells were stained with DioC6 and propidium iodide to evaluate cell viability and analyzed by Olympus IX83 inverted microscope (Olympus Europa, Hamburg, Germany) supplemented with a STEDYCON ultrawide extension platform with a magnification of 20 × 100. The fraction of non-viable cells was evaluated in each Z-stack by using BioFilmAnalyzer (version 1.2) software and the average percent is shown on each image.

**Figure 6 ijms-24-16090-f006:**
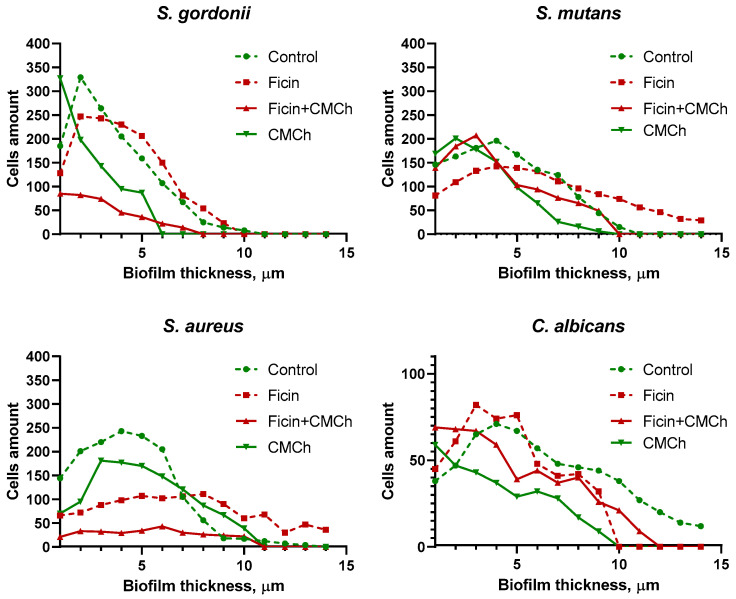
The distribution of total number of cells by biofilm layers obtained as Z-stacks by confocal laser scanning microscopy. Microbial 24 h-old biofilms were treated for 3 h with either Ficin, pure CMCh, or CMCh200-immobilized Ficin. Then, cells were stained with DioC6 and propidium iodide to evaluate cell viability and analyzed by an Olympus IX83 inverted microscope (Olympus Europa, Hamburg, Germany) supplemented with a STEDYCON ultrawide extension platform. The total number of cells was evaluated in each Z-stack by using BioFilmAnalyzer software.

**Figure 7 ijms-24-16090-f007:**
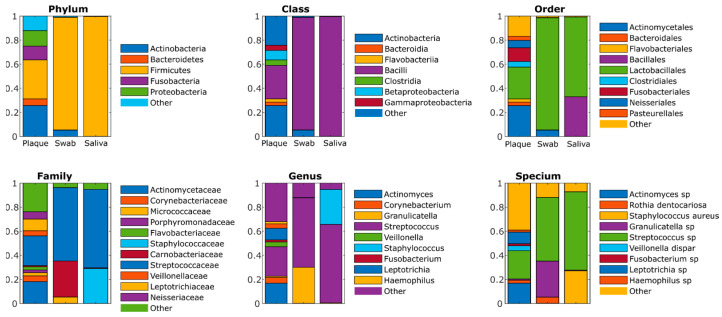
Comparison of the microbiota structure in tooth plaque and biofilms grown in vitro by inoculation of saliva and tooth swabs, respectively, obtained from four healthy individuals and averaged. The *Y*-axis shows the frequency of occurrence of representatives of the microbiota in tooth plaque and biofilms.

**Figure 8 ijms-24-16090-f008:**
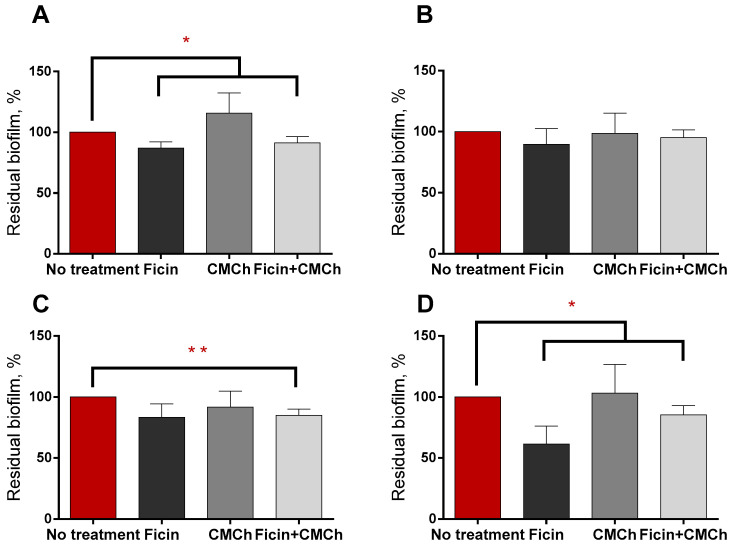
The effect of soluble and carboxymethyl chitosan-immobilized Ficin on the biofilms grown in vitro by inoculation of the tooth swab obtained from four apparently healthy volunteers (**A**–**D**). 24 h-old biofilms grown in BM broth were gently washed by PBS and loaded with fresh BM broth supplemented with either soluble Ficin (500 µg/mL) or CMCh-immobilized Ficin (35 mg/mL) followed by 3 h incubation and quantification of the residual biofilms by crystal-violet staining. Statistically significant differences at *p* < 0.05 are shown with single asterisks (*), while differences at *p* < 0.1 are shown with double asterisks (**). A mean value ± standard deviation from nine independent measurements is shown.

**Figure 9 ijms-24-16090-f009:**
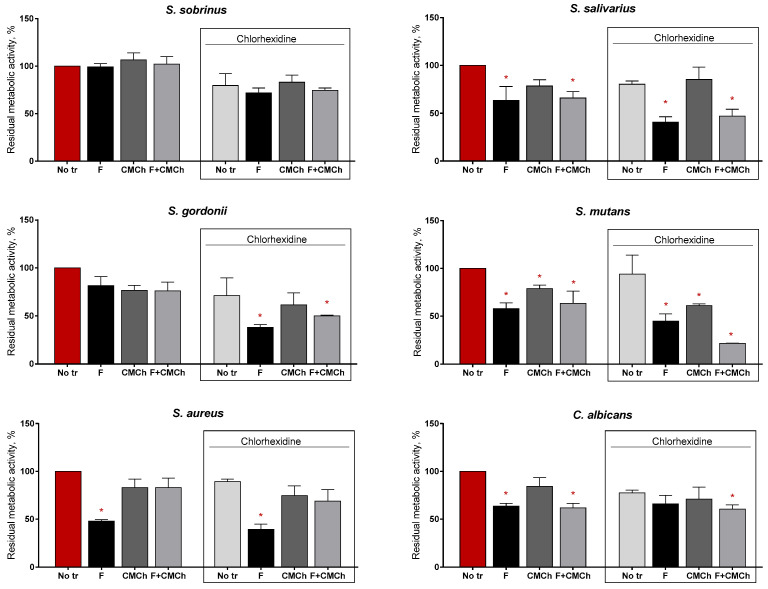
The effect of soluble and immobilized Ficin on the susceptibility of biofilms-embedded pathogens to antimicrobials. Either soluble or chitosan immobilized Ficin was added to the 24 h old biofilms until the final concentrations of 500 µg/mL and 35 mg/mL, respectively. Chitosan was added until 35 mg/mL. The final concentration of chlorhexidine was 16 μg/mL. After 24 h incubation, the biofilms were washed twice with sterile 0.9% NaCl. The adherent cells were scratched, resuspended and their viability was analyzed by MTT assay. Asterisks (*) denote the statistically significant difference in the residual metabolic activity of cells in the wells with treatment vs. control wells (*p* < 0.05). A mean value ± standard deviation from nine independent measurements is shown.

**Table 1 ijms-24-16090-t001:** The residual protein and enzymatic activity of Ficin after its immobilization on carboxymethyl chitosan (CMCh) with molecular weights of either 200, 350, or 600 kDa. The protein amount and activity in the Ficin solution used for immobilization (20 mg/mL) was considered as 100%. A mean value ± standard deviation from five independent measurements is shown.

Enzyme	Protein, mg/g of CMCh	Protein, % of Free Enzyme	Total Activity, U/mL	Total Activity, % of Free Enzyme	Optimal t, °C	Optimal pH
Soluble Ficin	20.0 ± 0.1	100	96 ± 2.2	100	37–60	7.5
Ficin on CMCh 200 kDa	9.7 ± 0.2	49 ± 1.1	14.9 ± 1.7	15	37–60	6.5–7.5
Ficin on CMCh 350 kDa	3.9 ± 1.2	19 ± 1.3	31.6 ± 1.4	33	37–60	6.5–7.5
Ficin on CMCh 600 kDa	6.4 ± 2.4	32 ± 0.7	62.2 ± 3.9	65	37–60	6.5–7.5

**Table 2 ijms-24-16090-t002:** The kinetic parameters of free and CMCh-immobilized Ficin. A mean value ± standard deviation of calculations from five independent measurements is shown.

Enzyme Form	K_m_, µM	V_max_, µM mg^−1^ min^−1^	k_cat_ (min^−1^)	V_max_/K_m_	k_cat_/K_m_
Free Ficin	20 ± 6.4	132 ± 20	6.3 ± 0.6	6.6	0.3
Immobilized on CMCh200	16 ± 4.5	19 ± 3	0.9 ± 0.1	1.2	0.1
Immobilized on CMCh350	17 ± 5.0	42 ± 5	2.0 ± 0.2	2.5	0.1
Immobilized on CMCh600	13 ± 2.9	78 ± 8	3.7 ± 0.2	6.0	0.2

**Table 3 ijms-24-16090-t003:** The distribution of the most abundant bacterial genus in the tooth plaque and in the biofilms grown in vitro by inoculation of saliva and tooth swabs, respectively.

Genus	Tooth Plaque	Biofilm Grown by Inoculation of Tooth Swab	Biofilm Grown by Inoculation of Saliva
*Actinomyces*	16.9%	0.2%	0.0%
*Corynebacterium*	4.7%	0.0%	0.0%
*Staphylococcus*	0.0%	0.0%	29.0%
*Granulicatella*	1.0%	29.9%	0.6%
*Streptococcus*	24.6%	57.6%	65.1%
*Veillonella*	3.9%	0.0%	0.0%
*Fusobacterium*	1.8%	0.0%	0.0%
*Leptotrichia*	9.7%	0.2%	0.0%
*Neisseria*	1.6%	0.0%	0.0%

**Table 4 ijms-24-16090-t004:** Biodiversity indices in the tooth plaque and the biofilms grown in vitro by inoculation of saliva and tooth swab, respectively.

Indices	Tooth Plaque	Biofilm Grown by Inoculation of Tooth Swab	Biofilm Grown by Inoculation of Saliva
PD_whole_tree	17.7	10.3	11.1
Chao1	644	325	257
Shennon	6.35	4.29	3.06
Simpson	0.96	0.84	0.73

**Table 5 ijms-24-16090-t005:** The ratio of the residual metabolic activity of the treated biofilms and CV-stain data.

Species	Ficin Treatment	CMCh Treatment	CMCh-Ficin Treatment
*S. aureus*	0.72	0.76	0.94
*C. albicans*	1.02	0.59	0.58
*S. gordonii*	0.84	1.35	1.14
*S. salivarius*	0.81	1.17	0.70
*S. mutans*	0.94	0.92	0.84
*S. sobrinus*	1.19	1.13	1.08

## Data Availability

The data presented in this study are available on request from the corresponding author.

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
