# Peer review of "The Effect of Ficin Immobilized on Carboxymethyl Chitosan on Biofilms of Oral Pathogens"

_ijms, 2023, doi:10.3390/ijms242216090_

Round 1
Reviewer 1 Report
Comments and Suggestions for Authors
Baidamshina et al., immobilized the ficin to carboxymethyl chitosan and compared it antibiofilm efficiency with free Ficin. Although authors claim the significant antibiofilm potential of presented material, additional evidence are required to support their conclusion. I have following comment to authors.
1. In introduction authors introduces the chlorhexidine which is often used to prevent or eradication of plaque. Chlorhexidine is used as mouth wash solution. How does the author see the potential application of Ficin immobilized CMCh? If it is supposed to be used as mouth wash solution; the antibiofilm efficiency should be evaluated after a couple of minutes of treatment with test agent.
2. It would be great if the author could show some electron microscopic images of Ficin immobilized CMCH.
3. The antibiofilm activity measurement may have been influenced by CHCH, since it is likely to settle down and crystal violet is a non-specific binder. Thus, it is likely that the effect may have been underestimated. The authors are requested to perform the viability assessment by CFUs counting.
4. How the concentration of free Ficin and CMCh embeded fincin was selected. It is hard to find a rational in comparing the results between two treatment agents with huge difference in concentration.
5. Figure 5: Authors are recommended perform the quantitative analysis of acquired confocal images to demonstrate the fold of decrease in biofilm biomass after the treatment. The obvious no difference in microbial viability must be supported by CFUs counting.
6. Page 8: Sample B and C in the text is confusing, author most elaborate what that denotes to.
Comments on the Quality of English LanguageProofreading of manuscript is required
Author Response
Dear Editor,
Please find enclosed the revised version of the paper “The Effect of Ficin Immobilized on Carboxymethyl Chitosan on Biofilms of Oral Pathogens” by Diana Baidamshina, Elena Trizna, Svetlana Goncharova, Andrey Sorokin, Maria Lavlinskaya, Anastasiya Melnik, Leysan Gafarova, Maya Kharitonova, Olga Ostolopovskaya, Valeriy Artyukhov, Evgenia A. Sokolova, Marina Holyavka, Mikhail Bogachev Airat R. Kayumov and Pavel V. Zelenikhin that we would like to resubmit to International Journal of Molecular Sciences.
We thank Reviewers for their positive evaluation of our manuscript. We have revised the paper accordingly and have tried to incorporate all suggestions raised by reviewer. Moreover, we added new data based on quantification of CLSM images and evaluation of viability decrease.
In the following, we response to particular concerns raised by the Reviewers point by point.
Reviewer’s #1 question:
- In introduction authors introduces the chlorhexidine which is often used to prevent or eradication of plaque. Chlorhexidine is used as mouth wash solution. How does the author see the potential application of Ficin immobilized CMCh? If it is supposed to be used as mouth wash solution; the antibiofilm efficiency should be evaluated after a couple of minutes of treatment with test agent.
Author’s response:
The potential application of Ficin immobilized CMCh could be different, from an additives to the mouth wash solutions (since the CMCh is soluble), toothpaste, gum to antiseptic films widely used in stomatology. While the exposition of plaque to Ficin in wash solutions, toothpaste, gum will be short, the application in antiseptic films could be combined with antiseptics and would have long-term contact with bacterial biofilms.
In the study, we have chosen 3 h exposition as a minimal time frame when we see statistically significant effects, while in real life conditions differ from in vitro ones and requires additional assays.
Reviewer’s #1 question:
- It would be great if the author could show some electron microscopic images of Ficin immobilized CMCH.
Author’s response:
We agree with a reviewer that microscopic images of Ficin immobilized CMCH seems to be required. For regret, they become uninformative, since the adhered protein makes their surface smooth and uniform, see image in file. Therefore we would not like to add them to the paper.
Reviewer’s #1 question:
- How the concentration of free Ficin and CMCh embeded ficin was selected. It is hard to find a rational in comparing the results between two treatment agents with huge difference in concentration.
Author’s response:
In previous experiments, 1000 ug/mL of soluble Ficin exhibited a biofilm eradicating activity (Baidamshina, SciRep 2017, IJBM 2020). In these experiments, we normalized the soluble and immobilized enzyme by total protein. In a large-scale immobilization of Ficin on CMCh200 a bound protein was 14.1 mg/g of СMCh, assuming that 70 mg/mL of CMCh200 immobilized Ficin corresponds to 1 mg/mL of pure enzyme by total protein. Since 70 mg/mL of CMCh200 gives a false negative results, we decreased its amount twice. Therefore, 35 mg/ml of immobilized Ficin corresponds to 500 µg/mL of the soluble enzyme. This explanation is added to the text
Reviewer’s #1 questions:
- The antibiofilm activity measurement may have been influenced by CHCH, since it is likely to settle down and crystal violet is a non-specific binder. Thus, it is likely that the effect may have been underestimated. The authors are requested to perform the viability assessment by CFUs counting.
- Figure 5: Authors are recommended perform the quantitative analysis of acquired confocal images to demonstrate the fold of decrease in biofilm biomass after the treatment.
- The obvious no difference in microbial viability must be supported by CFUs counting.
Author’s response:
We thank the reviewer for these important points. Since these questions are linked together, we would like to give a combined answer.
As suggested by the reviewer, we performed the quantification of cells amount (total, viable, dead) in each Z-stack of confocal images by using in-house developed software (BiofilmAnalyzer https://doi.org/10.1371/journal.pone.0193267). Data reflecting the total amount of cells per layer are shown in the new Figure 6. As could be seen from the figure, the cells amount significantly decreases in samples treated with both types of the enzyme, furthermore, the highest reduction in cells amount occurs after the treatment with CMCh-immobilyzed Ficin.
Furthermore, we counted the ratio of the viable/dead cells in each layer, and the average percentage of dead cells per layer (1 um) is shown on Figure 5. AS can be seen from these data, CMCh have a toxic effect only to C.albicans, we added this to the text. We suggest this counting since the CFUs counting will show false-positive results: with the biofilm destruction the amount of cell will decrease too leading to apparent cell death.
Finally, the decrease in biofilm biomass (Fig 1) fits well with changes in residual metabolic activity (Fig 9), that in turn is a proof of biofilm destruction. From the other hand, we calculated the ratio of residual metabolic activity of treated biofilms (Fig 9) and the CV-stain data (Fig 1), and the factor remains at about 0.7-1, suggesting that biofilm treatment rather decreases the biomass than kills the cells.
|
|
Ficin |
CMCh |
CMCh-Ficin |
|
S.aureus |
0,72 |
0,76 |
0,94 |
|
C.albicans |
1,02 |
0,59 |
0,58 |
|
S.gordonii |
0,84 |
1,35 |
1,14 |
|
S.salivarius |
0,81 |
1,17 |
0,70 |
|
S.mutans |
0,94 |
0,92 |
0,84 |
|
S.sobrinus |
1,19 |
1,13 |
1,08 |
Reviewer’s #1 question:
- Page 8: Sample B and C in the text is confusing, author most elaborate what that denotes to.
Author’s response:
Thank you for this attention, we made corrections.
Dr. Prof Airat Kayumov and Dr. Pavel Zelenikhin, for all authors

Reviewer 2 Report
Comments and Suggestions for Authors
This research is under the scope of this journal; the topic is relevant for readers, and this research deals with potentially significant knowledge to the field.
However, there are some concerns about the present manuscript:
Introduction
Opportunistic microorganisms can also cause systemic diseases, including diseases of the gastrointestinal tract and the cardiovascular system. Please consider:PMID: 37755045
Please add the null hypothesis in the aim section.
Material and Methods?
Number of samples in each experiment
- Sample size calculation is not clear. Please better describe the primary outcome utilized, standard deviation and the mean average among groups
Statistical Analysis?
Figures: please include the type of images, the used devices, and the magnifications - Lots of results sections include discussion parts, thus, please separate it - Figures contain different images which were taken with various microscopes. these microscopes were not clearly identified among the methods as well as the methods which were used to perform these images
There are many mistakes in the references section and in the text
The discussion is also misleading. What is the novelty of this paper???
Please add the limitations of the study in the discussion and provide an adequate debate. Discussion should be better organized.
Comments on the Quality of English LanguageI would suggest a double-check of the grammar and style of this paper writing. Ensure that your sentences are clear and concise. Also, make sure that your writing maintains a consistent tone throughout the section.
Author Response
Dear Editor,
Please find enclosed the revised version of the paper “The Effect of Ficin Immobilized on Carboxymethyl Chitosan on Biofilms of Oral Pathogens” by Diana Baidamshina, Elena Trizna, Svetlana Goncharova, Andrey Sorokin, Maria Lavlinskaya, Anastasiya Melnik, Leysan Gafarova, Maya Kharitonova, Olga Ostolopovskaya, Valeriy Artyukhov, Evgenia A. Sokolova, Marina Holyavka, Mikhail Bogachev Airat R. Kayumov and Pavel V. Zelenikhin that we would like to resubmit to International Journal of Molecular Sciences.
We thank Reviewers for their positive evaluation of our manuscript. We have revised the paper accordingly and have tried to incorporate all suggestions raised by reviewer. Moreover, we added new data based on quantification of CLSM images and evaluation of viability decrease.
In the following, we response to particular concerns raised by the Reviewers point by point.
Reviewer’s #2 question:
Introduction
Opportunistic microorganisms can also cause systemic diseases, including diseases of the gastrointestinal tract and the cardiovascular system. Please consider:PMID: 37755045
Author’s response:
We added this information to Introduction section. Nevertheless, the use of enzymes is limited to the topical application (skin, mucosa etc), and they can’t be used for systemic treatment. Threfore we focused on the oral microflora.
Reviewer’s #2 question:
Please add the null hypothesis in the aim section.
Author’s response:
Is added as suggested
Reviewer’s #2 question:
Material and Methods
Number of samples in each experiment
- Sample size calculation is not clear. Please better describe the primary outcome utilized, standard deviation and the mean average among groups
Statistical Analysis?
Author’s response:
The sample size is mentioned for each table and figure where applicable.
Reviewer’s #2 question:
Figures: please include the type of images, the used devices, and the magnifications - Lots of results sections include discussion parts, thus, please separate it - Figures contain different images which were taken with various microscopes. these microscopes were not clearly identified among the methods as well as the methods which were used to perform these images
Author’s response:
We have updated Methods section and mentioned microscopes in the figure captions
Reviewer’s #2 question:
There are many mistakes in the references section and in the text
Author’s response:
We have checked the list
Reviewer’s #2 question:
What is the novelty of this paper???
Author’s response:
Here we show the first example (to the best of our knowledge) of antibiofilm activity of proteases immobilized on soluble chitosan, since previously the immobilization was carried out on insoluble carriers.
Reviewer’s #2 question:
The discussion is also misleading.
Please add the limitations of the study in the discussion and provide an adequate debate. Discussion should be better organized.
Author’s response:
The Discussion has been revised
Reviewer’s #2 question:
Comments on the Quality of English Language
I would suggest a double-check of the grammar and style of this paper writing. Ensure that your sentences are clear and concise. Also, make sure that your writing maintains a consistent tone throughout the section.
Author’s response:
We have checked the Language
Dr. Prof Airat Kayumov and Dr. Pavel Zelenikhin, for all authors

Round 2
Reviewer 1 Report
Comments and Suggestions for Authors
Comments are well responsed by authors with some new analysis.
Reviewer 2 Report
Comments and Suggestions for Authors
Accept